# Terpene Glycosides from *Sanguisorba officinalis* and Their Anti-Inflammatory Effects

**DOI:** 10.3390/molecules24162906

**Published:** 2019-08-10

**Authors:** Da-Le Guo, Jin-Feng Chen, Lu Tan, Meng-Ying Jin, Feng Ju, Zhi-Xing Cao, Fang Deng, Li-Na Wang, Yu-Cheng Gu, Yun Deng

**Affiliations:** 1The Ministry of Education Key Laboratory of Standardization of Chinese Herbal Medicine, State Key Laboratory, Breeding Base of Systematic Research Development and Utilization of Chinese Medicine Resources, School of Pharmacy, Chengdu University of Traditional Chinese Medicine, Chengdu 611137, China; 2School of Nursing, Sichuan Tianyi College, Mianzhu 618200, China; 3Syngenta Jealott’s Hill International Research Centre, Berkshire RG42 6EY, UK

**Keywords:** terpene glycosides, structural elucidation, *Sanguisorba officinalis*, zebrafish, anti-inflammatory

## Abstract

Two new terpene glycosides (**1**–**2**) along with two known analogs (**3**–**4**) were obtained from the root of *Sanguisorba officinalis,* which is a common traditional Chinese medicine (TCM). Their structures were elucidated by nuclear magnetic resonance (NMR), electrospray ionization high resolution mass spectrometry (HRESIMS), and a hydrolysis reaction, as well as comparison of these data with the literature data. Compounds **1**–**4** exhibited anti-inflammatory properties *in vitro* by attenuating the production of inflammatory mediators, such as nitric oxide (NO) as well as tumor necrosis factor-α (TNF-α) and interleukin-6 (IL-6). An anti-inflammatory assay based on the zebrafish experimental platform indicated that compound **1** had good anti-inflammatory activity in vivo by not only regulating the distribution, but also by reducing the amount of the macrophages of the zebrafish exposed to copper sulfate.

## 1. Introduction

*Sanguisorba officinalis* belongs to the Rosaceae family and is distributed widely in Asia, Europe, North Africa, and North America [1]. Its dried root has been commonly used as a traditional Chinese medicine for the treatment of burn, scalds, inflammation, and hemorrhage by the Chinese over thousands of years [2] due to its astringent and analgesic properties [3]. Previous studies indicated that triterpenoids [4], triterpenoid glycosides [5], lignans [6], lignans glycosides [7], polysaccharides [8], hydrolyzable tannins [9], as well as terpene glycosides [10] could partially account for its therapeutic effects.

As a response of the immune system to irritation and infection, inflammation represents a key component in normal tissue homeostasis. Accordingly, deregulated inflammatory responses result in severely detrimental chronic conditions, including the accumulation of inflammatory mediators [11] as well as the infiltration of the affected tissue by leukocytes of the innate immune system [12]. Zebrafish (*Danio rerio*) has been extensively used to study the pathogenesis of human diseases in many fields [13]. Due to the similarity of the innate immune systems between zebrafish and mammals, as well as the transparency of the zebrafish larvae, which allows real-time observation of the inflammatory response of fluorescent macrophages [14], zebrafish inflammatory models can successfully recapitulate the infiltration process of leukocytes.

With the purpose of searching for substances with anti-inflammatory effect from *S. officinalis*, we continued our studies and four compounds, including two previously undescribed terpene glycosides (**1**–**2**) along with two analogs (**3**–**4**), were obtained from of the root of *S. officinalis* (Figure 1). Their structures were identified by one-dimension and two-dimension nuclear magnetic resonance (NMR), high-resolution electrospray ionization mass spectroscopy (HRESIMS) spectra, a hydrolysis reaction, as well as by comparison of these data with the literature. The anti-inflammatory effects of compounds **1**-**4** were showed that treating RAW264.7 cells resulted in decreased production of nitric oxide (NO), tumor necrosis factor-α (TNF-α) and interleukin-6 (IL-6) Besides, compound **1** showed anti-inflammatory effect in zebrafish inflammation model. The details of the isolation, structural elucidation and bio-activitiy measurements of these compounds are discussed below.

## 2. Results and Discussion

Compound **1** was obtained as a yellow powder, HRESIMS ion at *m*/*z* 507.1849 [M + Na]^+^ (Appendix A) indicated its molecular formula should be C_23_H_32_O_11_ (calculated for C_23_H_32_O_11_Na^+^, 507.1837). The ^1^H-NMR spectrum (Appendix A) displayed typical galloyl signals at *δ* 7.09 (2H, s, H-3″ and H-7″); two olefinic signals at *δ* 5.24 (1H, t, *J* = 6.5 Hz, H-2) and *δ* 5.23 (1H, t, *J* = 7.0 Hz, H-6); and two methyl signals at *δ* 1.75 (3H, s, H-8) and 1.60 (3H, s, H-10). The ^13^C-NMR and heteronuclear multiple-quantum coherence (HSQC) spectra of compound **1** (Appendix A) exhibited 21 carbon signals, including an ester carboxyl at *δ* 168.4 (C-1″); four olefinic carbons at *δ* 142.0 (C-3), 136.2 (C-7), 128.5 (C-6) and 121.5 (C-2), as well as four aromatic carbons at *δ* 146.5 (C-4″ and 6″), 139.9 (C-5″), 121.4 (C-2″), and 110.2 (C-3″ and C-7″); four oxymethine carbon signals at *δ* 78.4 (C-3′), 75.5 (C-5′), 75.0 (C-2′), and 71.9 (C-4′); three oxymethylene carbon signals at *δ* 66.2 (C-1), 65.1 (C-6′), and 61.4 (C-8); two methylene carbons at *δ* 41.7 (C-4) and 26.8 (C-5); two methyl signals at *δ* 21.7 (C-9) and 16.4 (C-10). An 8-hydroxygeraniol moiety [10] was established from the ^1^H-^1^H correlation spectroscopy (COSY) cross signals as shown in Figure 2 for the H-1′/H-2′, H-3′/H-4′ and H-4′/H-5′ coupling systems, as well as the key heteronuclear mutiple-bond correlations (HMBC) of H-10/C-2, C-3, C-4 and H-9/C-6, C-7, C-8. Its configuration was confirmed by the nuclear Overhauser effect (NOE) correlations (Figure 2) between H-1 and H-10, as well as between H-6 and H-9; the presences of a glucopyranosyl moiety was deduced from the ^1^H-^1^H COSY cross signals for the H-1′/H-2′/H-3′/H-4′/H-5′/H-6′ coupling systems and the key HMBC correlations H-1′/C-3′, C-5′ and H-6′/C-4′ and the larger coupling constants (7.8 Hz) of the anomeric proton at *δ*_H_ 4.31 indicated a *β*-configuration of the glucosyl moiety. A galloyl group was further established from the HMBC cross signals for the H-3″, 7″/C-1″, C-2″, C-5″, C-4″, 6″. These three moieties were linked together by the key HMBC correlations of H-1/C-1′ and H-6′/C-1″ and generated the structure of compound **1**. The sugar component was identified as d-glucose by HPLC analysis after acid hydrolysis and derivatization of **1** combined with optical rotation comparison. Finally, the structure of compound **1** was confirmed to be 8-hydroxygeraniol-1-*O*-(6-*O*-galloyl)-*β*-d-glucopyranoside.

Compound **2** was isolated as a white powder, the molecular formula of **2** was assigned as C_21_H_36_O_11_ based on its HRESIMS (Appendix A) ion peak as *m*/*z*: 487.2167 (calculated for C_21_H_36_O_11_Na, 487.2150). Interpretation of the ^1^H- and ^13^C-NMR data (Appendix A)of compound **2** indicated that the terpene fragment of this compound was the same as in **1**, but the sugar moiety was substantially different. The coupling constants of the anomeric proton of glucose at *δ* 4.35 (d, 1H, *J* = 7.3 Hz) and of arabinose at *δ* 5.04 (d, 1H, *J* = 1.1 Hz) indicated that the glucose moiety was in a *β*-configuration and the arabinose moiety was in an *α*-configuration. Acid hydrolysis of **2** with 2 mol/L CF_3_COOH (TFA) afforded the sugar a d-glucose and an l-arabinose, which was identified by HPLC analyses after acid hydrolysis and derivatization. Based on the HMBC correlations (Figure 2) between H-1 (*δ* 4.25–4.41) and C-1′ (*δ* 102.3), as well as between H-1″ (*δ* 5.04) and C-6′ (*δ* 68.3), the structure of **2** was elucidated as 8-hydroxygeraniol-1-*O*-*α*-l-arabinofuranosyl-(1→6)-*β*-d-glucopyranoside.

The known analogs were identified as 8-hydroxygeraniol-*β*-d-glucopyranoside (**3**) [15] and (2*E*)-7-hydroxy-3,7-dimethyl-2-octenyl-6-*O*-*a*-l-arabinofuranosyl-*β*-d-glucopyranoside (**4**) [16] by comparison with those data from the literature.

The in vitro anti-inflammatory activities of compounds **1**–**4** were studied on LPS-stimulated macrophages. The effects of the isolated compounds on the viability of RAW264.7 cells were evaluated by MTT firstly. Treatment with the test samples for 48 h at the concentrations of 0, 3.75, 7.5, 15, 30, 60, 120 μg/mL and all four compounds at a concentration of 60 μg/mL produced no significant cytotoxic effects (Appendix A).

Our in vitro tests indicated that compounds **1**–**4** could attenuate LPS-induced NO production in RAW264.7 macrophages. As shown in Figure 3, the level of NO production was significantly increased in the LPS-treated cells compared with the untreated cells. However, NO production was significantly reduced in the cells pretreated with compounds **1**–**4** in a concentration-dependent manner. Besides, compounds **1**–**4** also could suppress the production of proinflammatory cytokines such as IL-6 and TNF-α. As we can see from Figure 4, compared with the vehicle treatment, LPS induced a significant increase in IL-6 and TNF-α. In contrast, compounds **1**–**4** at 15 and 60 μg/mL significantly reduced the production of IL-6, and compounds **1** and **3** at 15 and 60 μg/mL reduced the production of TNF-α.

Noninvasive damage to lateral line neuromast cells can induce a robust acute inflammatory response. Exposure of fish larvae to sublethal concentrations of copper sulfate selectively damages the sensory hair cell population, inducing the infiltration of macrophages to neuromasts, where inflammation can be assayed in real-time using transgenic fish expressing fluorescent proteins in leukocytes [13]. To test the in vivo anti-inflammatory activities of these four isolated compounds, a transgenic zebrafish line Tg (mpx: GFP) was induced by a copper sulfate solution and the migration of green fluorescent protein (GFP)-labeled macrophages was observed. Our study showed that compound **1** could significantly inhibit the migration of macrophages around the neuromast at the dose of 15 μg/mL. Besides, there were no significant differences in the amount of macrophages between zebrafish larvae in the blank group and the zebrafish larvae incubated with compound **1** at the dose of 60 μg/mL following exposure to 10 μM copper sulfate, which indicated that compound **1** had anti-inflammatory activity (Figure 5).

## 3. Materials and Methods 

### 3.1. General Experimental Procedures

Optical rotations were determined on a Perkin-Elmer-241 polarimeter (Perkin Elmer, Inc., Waltham, MA, USA) at room temperature. Ultraviolet-visble (UV) absroption spectra were recorded on a Perkin-Elmer Lambda 35 UV-VIS spectrophotometer (Perkin Elmer, Inc., Waltham, MA, USA). Infrared (IR) spectra were measured by the Perkin-Elmer one FT-IR spectrometer (KBr) (Perkin Elmer, Inc., Waltham, MA, USA). 1D and 2D NMR were carried out on a Bruker-Ascend-400 MHz instrument (Bruker, Bremen, Germany) at 300 K, with TMS as the internal standard. HRESIMS were measured using a Synapt G2 HDMS instrument (Waters Corporation Milford, MA, USA). The preparative HPLC was performed on a NP7000 serials pump (Hanbon Sci. & Tech., Jiangsu, China) equipped with a Kromasil ODS column (10 × 250 mm, 5 μm, Akzo Nobel Pulp and Performance Chemicals AB, Bohus, Sweden) using a NU3000 serials UV detector (Hanbon Sci. & Tech., Jiangsu, China). The analytic HPLC was performed on an Utimate 3000 serials pump (Thermo Scientific, Waltham, MA USA) equipped with a Luna NH2 column (4.6 × 250 mm, 5 μm, Phenomenex, Inc. Torrance, USA) using a ELSD6000 serials ELSD detector (Alltech Technology Ltd., New Westminster, BC, USA) or a Kromasil Eternity XT-5-C18 column (4.6 × 250 mm, 5 μm, Akzo Nobel Pulp and Performance Chemicals AB, Bohus, Sweden) using a Utimate 3000 DAD detector (Thermo Scientific, Waltham, MA, USA). The column chromatography (CC) was performed with silica gel (200–300 mesh, Qingdao Haiyang Chemical Co., Qingdao, China) and Sephadex LH-20 (GE-Healthcare Bio-Sciences AB, Uppsala, Sweden). The zebrafish were bred in a zebrafish breeding system (Beijing Aisheng Technology Development Co., Ltd., Beiiing, China). The zebrafish embryos and larva were incubated in an MGC-100 constant temperature incubator (Shanghai Yiheng Scientific Instrument Co., Ltd.). The zebrafish were immobilized by sodium carboxymethyl cellulose (CMC-Na, batch number: M0202A, Dalian Meilun Biotechnology Co., Ltd.) and observed by an M165-FC type fluorescence microscopy imaging system (Leica, Germany). The NO assay kit was purchased from Beyotime Biotechnology Co, Ltd. (Shanghai, China, Batch No: 032519190612). The mouse enzyme-linked immunosorbent assay (ELISA) kits were purchased from NeoBioscience Technology Co, Ltd. (Shenzhen, China, Batch No: M190726-004a, M190726-102a). The lipopolysaccharide was purchased from Beijing Solarbio Science & Technology Co., Ltd. (Beijing, China). The thiazolyl blue was purchased from Sigma-Aldrich (USA). The L-arabinose and D-glucose were purchased from Energy Chemical (Shanghai, China). The L-cysteine methyl ester hydrochloride was purchased from Chroma-Biotechnology Co. Ltd (Chengdu, China). The phenylisothiocyanate was purchased from Aladdin (Shanghai, China). All solvents used were of analytical grade.

### 3.2. Plant Material

*S**. officinalis* were provided by Diao Group Tianfu Pharmaceutical Group co., LTD, Sichuan province, China, and taxonomically identified by its pharmacist, Mr Qiu He. A voucher specimen (access number: 20150920) was deposited at the ministry of education key laboratory of standardization of Chinese herbal medicine, Chengdu University of TCM.

### 3.3. Extraction and Isolation

The air-dried and powdered of *S. officinalis* (9.97 kg) were extracted with 70% EtOH (10 l, 3 times) under reflux. The extract (3.31 kg) was suspended in water and partitioned with EtOAc and then with n-BuOH. The n-BuOH extract (1.47 kg) was subjected to D101 macroporous resin CC (10 × 120 cm), using a stepwise elution with ethanol/water (0%, 30%, 50%, 70%, 95%; *v*/*v*; 4L for each steps) to give five fractions. The fraction of 30% ethanol (0.74 Kg) was subjected to a column of HP-20 CC (6 × 50 cm) eluted with ethanol/water (0%, 10%, 20%, 30%, 40%, 50%, 100%, *v*/*v*; 900 mL each gradient) to yield seven fractions, F2a–F2g. Fraction F2d–F2f (84.6 g) was subsequently subjected to Toyopearl HW-40 CC (4 × 40 cm) eluted with ethanol/water (6:4, *v*/*v*) to afford three fractions, F1–F3. F3 (9.4 g) was subjected to a Sephadex LH-20 CC (150 cm L × 2 cm D) eluted with MeOH/H_2_O (1:1, *v*/*v*) to afford four subfractions, F3a–F3d. F3c (1.87 g) was purified by a preparative HPLC equipped with a Kromasil RP-C18 column (10 × 250 mm; 210 nm) to afford **1** (25.0 mg; MeOH/H_2_O: 38:62, *v*/*v*; 4 mL/min; t_R_: 42 min), **2** (16.9 mg; MeOH/H_2_O: 70:30, *v*/*v*; 4 mL/min; t_R_: 54 min), **3** (52.8 mg; MeOH/H_2_O: 40:60, *v*/*v*; 4 mL/min; t_R_: 18 min) and **4** (16.9 mg; MeOH/H_2_O: 30:70, *v*/*v*; 4 mL/min; t_R_: 68 min). 

8-hydroxygeraniol-1-*O*-(6-*O*-galloyl)-*β*-d-glucopyranoside (**1**). Yellow powder, [α]${}_{D}^{20}$ −30.6 (c 0.01, MeOH); IR (KBr) ν_max_ 3421.6, 2923.9, 1683.8, 1605.8, 1447.3, 1050.6 cm^−1^; UV λ_max_ 208.3 (3.82), 228.4 (3.80), 300.8 (3.37) nm; ^1^H-NMR and ^13^C-NMR spectral data which were unambiguously assigned by ^1^H-^1^H COSY, HSQC, and HMBC experiments (Appendix A) see Table 1; HRESIMS: *m*/*z* 507.1849 [M + Na]^+^ (C_23_H_32_O_11_Na^+^, 507.1837).

8-hydroxygeraniol-1-*O*-*α*-l-arabinofuranosyl-(1→6)-*β*-d-glucopyranoside (**2**). Yellow powder, [α]${}_{D}^{20}$ −95.1 (c 0.01, MeOH); IR (KBr) ν_max_ 3327.8, 2881.5, 2327.3, 1436.1, 1039.5 cm^−1^; UV λ_max_ 208.3 (3.64) nm; ^1^H-NMR and ^13^C-NMR spectral data which were unambiguously assigned by ^1^H-^1^H COSY, HSQC, and HMBC experiments (Appendix A) see Table 2; HRESIMS: *m*/*z* 487.2167 [M+Na]^+^ (calculated for C_15_H_24_O_2_Na, 487.2150).

### 3.4. Acid Hydrolysis and Derivatization of **1**–**2**

Compounds **1** (1.12 mg) and **2** (1.35 mg) were heated with trifluoroacetic acid (TFA) (2 M) for 6 h at 105 °C. The mixture was cooled and partitioned between CH_2_Cl_2_ (2 mL) and H_2_O three times. The aqueous phase firstly was evaporated to dryness under a vacuum and then analyzed by an HPLC (85 % CH_3_CN/H_2_O, flow rate = 1 mL/min) equipped with an evaporative light-scattering detector (ELSD) using a NH_2_ column, and the type of monosaccharides was grossly confirmed by comparing the retention time (Appendix A) with those of d-glucose (t_R_ =11.39 min) and l-arabinose (t_R_ = 7.89 min). At last, l-cysteine methyl ester hydrochloride (500 mg) and the residue of the aqueous phase were dissolved in pyridine (5 mL) and heated at 60 °C for 1 h, and after phenylisothiocyanate (0.5 mL) was added to the mixture and heated at 60 °C for 1 h. After evaporation of the solvent by rotary evaporator, the residue was further analyzed by HPLC (25% CH_3_CN/H_2_O, flow rate = 0.8 mL/min) equipped with a diode array detector (DAD) detector (under 250 nm) using a C-18 column. Finally, the absolute configuration of the monosaccharides was confirmed by comparing the retention time (Appendix A) with those of d-glucose (t_R_ = 14.03 min) and l-arabinose (t_R_ = 16.38 min).

### 3.5. Cell Viability Assay 

MTT colorimetric assays were performed to estimate the cytotoxic effects of compounds **1**–**4** on RAW264.7 macrophage cells. Firstly, the cells were seeded into 96-well plates at a density of 1 × 10^5^/mL cells per well. Various concentrations of compounds **1**–**4** (15, 30, 60, and 120 μg/mL) were added into the cultured cells. Then, the medium was replaced by 100 μL of fresh media containing MTT (0.5 mg/mL) 48 h after treatment with compounds **1**–**4**. After 3 h of incubation, the medium was discarded and dimethyl sulfoxide (DMSO) was added to dissolve the formazan crystals. The absorbance was measured at 570 nm to evaluate the inhibition effects of the tested compounds on the cell growth. 

### 3.6. Quantification of NO Production 

A Griess reaction was applied to measure the NO production by measuring the accumulation of nitrite. Briefly, Compounds **1**–**4** at 15, 30, and 60 μg/mL was cultured with RAW264.7 macrophages (1 × 10^5^/mL) receiving lipopolysaccharide (LPS) for 24 h. 100 μL of the Griess reagent was mixed with 100 μL of culture supernatant from each medium in a 96-well plate, After 15 minutes incubation at room temperature. A varioskan flash-3001 microplate reader (Thermo scientific, Waltham, MA, USA) was used to read the spectrophotometric absorbance at 550 nm wavelength. A cell-free medium without nitrite served as a blank control, and sodium nitrite was used as a standard for the calculation of nitrite concentration in the medium.

### 3.7. Measurement of Pro-Inflammatory Cytokine Production

A cytokine assay was conducted to examine the inhibitory action of compounds **1**–**4** on pro-inflammatory cytokines (IL-6, and TNF-α) as previously described. A mouse enzyme-linked immunosorbent assay (ELISA) kit (NeoBioscience Technology Co, Ltd. Shenzhen, China; Batch No M190726-004a, M190726-102a) was used to measure the proinflammatory cytokine production of supernatants from LPS (100 ng/mL) treated RAW264.7 macrophages.

### 3.8. Zebrafish Culture

The transgenic zebrafish line TG (mpx: GFP) were supported independently by the zebrafish experimental platform maintained at 28.5 °C water (pH 7.2–7.5; conductivity 500–550 μs/cm) under a 14 h light/10 h dark cycle. Four- to eight-month-old healthy zebrafish were paired to obtain zebrafish embryos and larvae. The zebrafish experimental operations were conducted according to the national institutes of health guide for the use and care of experimental animals and were approved by the animal experimentation ethics committee of the Chengdu University of TCM.

### 3.9. Anti-Inflammatory Assay Based on Zebrafish Experimental Platform 

The anti-inflammatory assay was carried out with dexamethasone (Appendix A) as the positive control. In general, healthy 3 dpf (3 days post-fertilization) zebrafish larvae were harvested and pro-treated with compounds or dexamethasone in different concentrations. After incubation for 1 h, the larvae were exposed to 10 μM copper sulfate under dark conditions in about 50 minutes to induce inflammation. Compared with the blank control group (only treated with 0.1% DMSO) as well as the model group (only treated with 10 μM copper sulfate), zebrafish larvae were anesthetized with tricaine and immobilized on slide with 1% CMC-Na and a fluorescence microscope and a couple-charged device camera was used to capture images of the migration of immune cells during the progression of inflammation.

## 4. Conclusions

Two new terpene glycosides along with two known analogs had been isolated from the roots of *S. officinalis*. The structures of new compounds were elucidated using NMR and HRESIMS, as well as a hydrolysis reaction. Compounds **1**–**4** exhibited anti-inflammatory properties in vitro by attenuating LPS-induced NO production, as well as reducing LPS-induced production of IL-6, and TNF-α. Compared with compounds **2**–**4**, compound **1** also showed an anti-inflammatory effect in vivo, which indicated that the galloyl group might be a key bioactive group for this kind of terpene glycoside to regulate macrophage distribution in zebrafish.

## Figures and Tables

**Figure 1 molecules-24-02906-f001:**
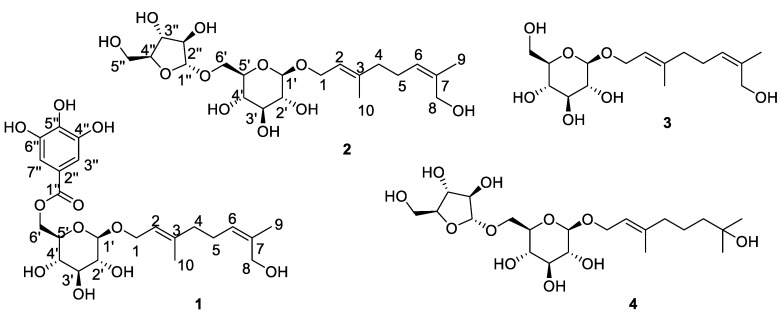
The structures of compounds **1**–**4** isolated from *Sanguisorba officinalis*.

**Figure 2 molecules-24-02906-f002:**
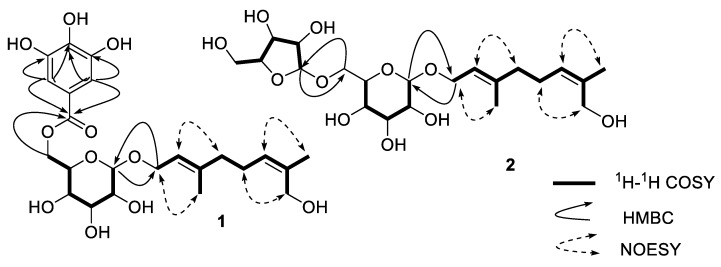
The ^1^H-^1^H COSY, key HMBC and key NOESY correlations of **1**–**2**.

**Figure 3 molecules-24-02906-f003:**
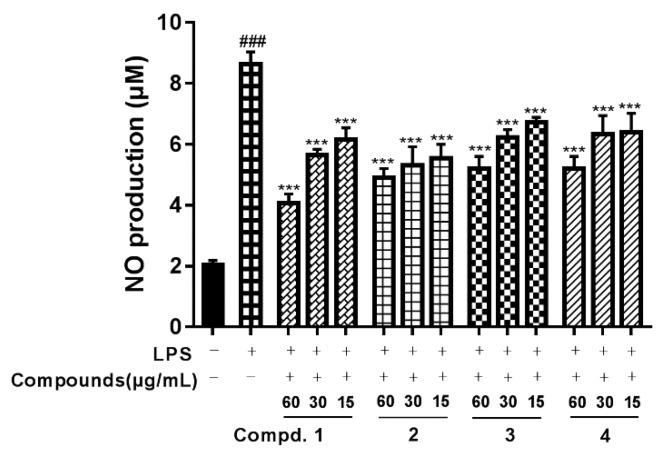
Effect of compounds **1**–**4** on NO production in LPS-stimulated macrophages. Macrophages were stimulated with LPS (100 ng/mL) for 24 h in the absence or presence of compounds **1**–**4** (15, 30 or 60 μg/mL). The production of NO was assayed in the culture media. Each value represents the mean ± standard error of the mean (SEM) (^###^
*p* < 0.001 versus with control group; *** *p* < 0.001 versus with LPS).

**Figure 4 molecules-24-02906-f004:**
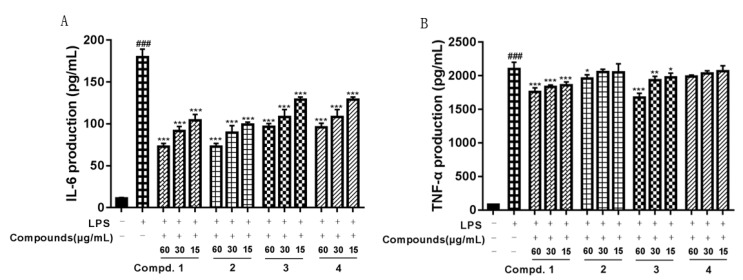
Effect of compounds **1**–**4** on production of IL-6 and TNF-α in LPS-stimulated macrophages. Macrophages were stimulated with LPS (100 ng/mL) for 24 h in the absence or presence of compounds **1**–**4** (15, 30 or 60 μg/mL). Production of IL-6 and TNF-α was assayed in the culture media using an enzyme-linked immunosorbent assay (ELISA) kit (**A**,**B**). Each value represents the mean ± SEM from three independent experiments. (^###^
*p* < 0.001 versus with control group, * *p* < 0.05; ** *p* < 0.01; *** *p* < 0.001 versus with LPS group).

**Figure 5 molecules-24-02906-f005:**
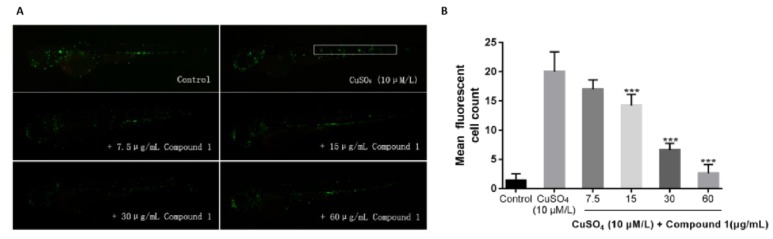
Effect of compound **1** on immune cell migration in CuSO_4_-stimulated inflammation in zebrafish. (**A**) Representative images of zebrafish treated with compound 1 and CuSO_4_ (**B**) Macrophages in the region of the neuromast were quantitatively analyzed. Each value represents the mean ± SEM from three independent experiments (*** *p <* 0.001 versus with the CuSO_4_ group).

**Table 1 molecules-24-02906-t001:** ^1^H- and ^13^C-NMR spectral data of **1**^a.^

Position	*δ*_H_ (*J* in Hz)	*δ* _C_	Position	*δ*_H_ (*J* in Hz)	*δ* _C_
1	4.18–4.28, m	66.2	2′	3.19–3.25, m	75.0
2	5.34, t (6.5)	121.5	3′	3.34–3.42, m	78.4
3	-	142.0	4′	3.35–3.45, m	71.5
4	2.03, t (7.5)	40.7	5′	3.49–3.56, m	75.5
5	2.16, dd (14.9, 7.4)	26.8	6′	4.55, dd (11.8, 2.0) 4.40, dd (11.8, 6.1)	65.1
6	5.23, t (7.0)	128.5	1″	-	168.4
7	-	136.2	2″	-	121.4
8	4.05, s	61.4	3″, 7″	7.09, s	110.2
9	1.75, s	21.7	4″, 6″	-	146.5
10	1.60, s	16.4	5″	-	139.9
1′	4.31, d (7.8)	102.6			

^a^ 400 MHz for ^1^H and 100 MHz for ^13^C in CD_3_OD.

**Table 2 molecules-24-02906-t002:** ^1^H- and ^13^C-NMR spectral data of **2**^a^.

Position	*δ*_H_ (*J* in Hz)	*δ* _C_	Position	*δ*_H_ (*J* in Hz)	*δ* _C_
1	4.25–4.41, m	66.4	2′	3.24, dd (8.8, 8.0)	75.0
2	5.46, m	121.8	3′	3.34–3.42, m	78.0
3	-	141.9	4′	3.30–3.40, d (8.9)	72.0
4	2.13, dd (14.4, 7.9)	40.8	5′	3.43–3.50, m	76.7
5	2.25, dd (14.7, 7.9)	26.7	6′	3.67, dd (11.1, 6.0) 4.04, dd (11.1, 2.3)	68.3
6	5.34, t (7.0)	128.1	1″	5.04, d (1.1)	110.1
7	-	136.0	2″	4.02–4.08, m	83.2
8	4.14, s	61.3	3″	3.89, dd (5.9, 3.2)	78.7
9	1.75, s	21.5	4″	4.01–4.07, m	85.9
10	1.60, s	16.4	5″	3.81, dd (11.9, 3.7) 3.67, dd (11.9, 5.3)	63.0
1′	4.35, d (7.3)	102.3			

^a^ 400 MHz for ^1^H and 100 MHz for ^13^C in CD_3_OD.

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
