# Peer review of "Terpene Glycosides from Sanguisorba officinalis and Their Anti-Inflammatory Effects"

_molecules, 2019, doi:10.3390/molecules24162906_

Round 1

Reviewer 1 Report

I send you my opinion about the article “Terpene glycosides from Sanguisorba officinalis and their anti-inflammatory effects” by Guo et al. and I find it quite interesting and I think that the manuscript is suitable for publication on Molecules after revision.

I would like to note some points that should be addressed:

-the results about compound 1 should be rewrite correctly, I signed many mistakes on the manuscript (see attachment). The chemical shift of C-4’ carbon could be at d 71.9 and of C-5’ carbon 75.0 (change in the table and text). The H-5’ proton is centered at d 3.53 as deduced by COSY and proton spectra.  

-Figure 1 please change the numeration on hydroxygeranoil moiety (C-8 should be CH2OH group)

-Figure 3 could be eliminated, it isn’t necessary.

Some mistakes and suggestions are indicated on the paper.

In its current state, it is suitable for publication on Molecules after revision.

Author Response

Thank you for your comments which are very helpful for revising and improving our paper. We have studied your comments carefully and have made revision which marked in red in the paper which we hope meet with approval. The responds to your comments are as following:

the results about compound 1 should be rewrite correctly, I signed many mistakes on the manuscript (see attachment). The chemical shift of C-4’ carbon could be at δ 71.9 and of C-5’ carbon 75.0 (change in the table and text). The H-5’ proton is centered at δ 3.53 as deduced by COSY and proton spectra.

Response: Thank you for your professional comments. We checked the NMR data again. The NMR data of the glucosyl moiety were reviesd. As we can see from the HSQC spectrum, the chemical shift of C-4’ could be at δ 71.5 and of C-5’ carbon 75.5. These mistakes were revised.

-Figure 1 please change the numeration on hydroxygeranoil moiety (C-8 should be CH2OH group)

Response: Thank you for your professional comments. The numeration on hydroxygeraniol was changed and the relative data were revised.

-Figure 3 could be eliminated, it isn’t necessary.

Response: Following the your suggestion, Figure 3 was eliminated and key NOESY correlations were exhibited in figure 2.

Some mistakes and suggestions are indicated on the paper.

Response: We appreciate your warm work earnestly. As per your suggestion, we checked this manuscript again and mistakes indicated on the paper were revised.

In its current state, it is suitable for publication on Molecules after revision.

Response: Thank you very much for your comments and suggestion.

Reviewer 2 Report

This is not a data-rich manuscript. Only two new phyto-chemicals are reported. The anti-inflammatory activities were not examined properly. What is the anti-inflammatory mechanism? What is the effective concentration? Zebra fish is not a well-recognized model. To assess the effective concentration precisely, cell culture model is more appropriate. Without new anti-inflammatory data obtained with other cell culture model, this manuscript should not be accepted.

Author Response

Thank you for your comments which are very helpful for revising and improving our paper. We have studied your comments carefully and have made revision which marked in red in the paper which we hope meet with approval. The responds to your comments are as following:

This is not a data-rich manuscript. Only two new phyto-chemicals are reported. The anti-inflammatory activities were not examined properly. What is the anti-inflammatory mechanism? What is the effective concentration? Zebra fish is not a well-recognized model. To assess the effective concentration precisely, cell culture model is more appropriate. Without new anti-inflammatory data obtained with other cell culture model, this manuscript should not be accepted.

Response: Following the your suggestion, The in vitro anti-inflammatory activities of compounds 1-4 were studied on LPS-stimulated macrophages. Firstly, the effects of isolated compounds on viability of RAW 264.7 cells were evaluated by MTT and compounds 1-4 at the concentrations of 60 μg/mL produced no signifificant cytotoxic effects. After then, the inhibitory activities of compounds 1-4 against nitric oxide (NO), as well as tumor necrosis factor-α (TNF-α) and interleukin-6 (IL-6) production in RAW 264.7 macrophages were examined. The results indicated that compounds 1-4 also exhibited anti-inflamatory in cell culture model via inhibiting the production of NO, TNF-α, and IL-6.

   We also thank you for your comments about Zebra fish model. We choose zebrafish as a model system to study anti-inflammatory effects of these compounds according to a paper titled “A high-throughput chemically induced inflammation assay in zebrafish”(d'Alencon, C.A. et al., BMC Biology. 2010, 8:151). it expounded the anti-inflammatory mechanism. Briefly, deregulated inflammatory reactions result in severely detrimental chronic conditions, and one of the hallmarks of the innate inflammatory response is infiltration of the affected tissue by leukocytes of the innate immune system (granulocytes and macrophages). Specific, noninvasive damage to lateral line neuromast cells can induce a robust acute inflammatory response. Exposure of fish larvae to sublethal concentrations of copper sulfate selectively damages the sensory hair cell population inducing infiltration of leukocytes (such as granulocytes and macrophages) to neuromasts. Inflammation can be assayed in real time using transgenic fish expressing fluorescent proteins in leukocytes (Green fluorescent protein (GFP)-labeled macrophages in our paper) or by histochemical assays in fixed larvae. The usefulness of this method for chemical and genetic screens to detect the effect of immunomodulatory compounds and mutations affecting the leukocyte response also had been demonstrated by d'Alencon, C.A. and his/her colleagues. We also introduced the anti-inflammatory mechanism briefly in page 1, lines 37-40 and page 4, lines 127-131. Meanwhile, zebrafish model can reflect the interaction between injured tissue and inflammatory system, and has certain advantages in studing the effect of drugs on the whole life system.

   Due to the fact that it is possible to study immunity by following the behavior of infiltrating cells in the living ransgenic fish expressing fluorescent proteins in leukocytes. Effective concentration was confirmed according to the number of macrophages around neuromasts between test groups stimulated with copper sulfate (10 μM) under dark condition for about 50 minutes in the absence or presence of compounds 1-4.

   Finally, We appreciate your warm work earnestly, and hope that the correction will meet with approval. Once again, thank you very much for your comments and suggestion!

Round 2

Reviewer 2 Report

The authors have added various new results in the manuscript. The manuscript appears to be acceptable.